# Abstract

**Scope of Reproducibility**

In this study, we evaluate the paper 'Identifying Through Flows for Recovering Latent Representations'. Specifically, we evaluate the papers' claimed practical advantages and effectiveness of their proposed method iFlow over previous methods, namely iVAE.

**Methodology**

First, we reproduce the obtained MCC scores for both iFlow and iVAE using the original code-base. To place these results into context, we also evaluate two baseline Flow models. Furthermore, we discuss the proposed method's usability, and apply it on a different Flow model, which is trained on the Half Moon dataset to analyse the learned latent representation. With this, we assess the benefit of using the proposed method over regular Flow. It takes around 20 minutes for an iFlow model, and 75 seconds for an iVAE model to train and evaluate conform to the original implementation and dataset on an RTX 2080 Ti GPU with 11GB of VRAM. Additionally, the iFlow network with planar-flow Flow model takes around 7 minutes to train on the same hardware. Finally, the two Flow models on the half moon datasets are trained on an AMD 3900X CPU with 32GB of DDR4 RAM, each taking roughly 3 minutes.

**Results**

Our results are within 2.5% of the values presented by the paper, verifying the authors' claim of iFlow's theoretical advancements over iVAE. However, when compared to the baseline Flow models, iFlow only shows up to 10% improvement in MCC scores, compared to a 45% improvement over iVAE. Furthermore, when analysing the learned latent representation for the Half Moon dataset iFlow does learn a more robust latent representation compared to Flow, and unlike Flow, is sometimes able to reach principled disentanglement, partly verifying the paper's claim of iFlow's practical advantages and effectiveness.

**What was easy**

The original code implementation was not difficult to setup and run specifically for the iFlow model. The code provided the proper run script for training and evaluating iFlow. Furthermore, implementing the proposed identifiability method to different flow models is not difficult - the authors provide a clear derivation of the objective function. Finally, in the code, a different use of the activation for the natural parameters is suggested, which we found to be straightforward to implement.

**What was difficult**

The code-base lacked documentation, thus besides running the default iFlow setup, running different models such as iVAE was quite challenging. In general, understanding the code itself, particularly the code used to generate the dataset, was not straightforward. No code was offered to save the results and construct the figures from the paper. Finally, despite the supposed support for using planar-flow instead of the default cubic spline-flow in the code base, training iFlow with planar-flow was not trivial. This was due to both an incorrect initialization of the planar-flow model, where it called the wrong class, and an incorrect return statement.

**Communication with original authors**

There has been no communication with the authors.

**Anonymous Author(s)**
Affiliation
Address
email

# 1   Introduction

One of the most fundamental goals of unsupervised representation learning is recovering the true joint distribution over the observed data and latent variables from which the data is generated. If we are able to learn this true distribution, we would also recover the true distribution over the latent variables. Since by definition latent variables are not observed, this is generally considered to be an extremely difficult task. However, it is shown that for a broad family of deep-latent-variable models recovering the true latent representation is possible [5]. Specifically, it is possible to recover the true latent representation if the model is *identifiable*. In broad terms, a model is identifiable if and only if it has a unique solution for a given set of parameters. By recovering the true latent representations, models are able to achieve principled disentanglement.

To build a deep latent-variable model which allows for recovery of true latent representations the authors of [7]. propose unifying identifiability with normalizing flows. Their proposed method iFlow is mainly built upon the foundation of the previous identifiable Variational Auto Encoder (iVAE) introduced by [5]. The authors of [7]. claim that flow-based models are particularly suitable for identifiable models, as the objective directly maximises the density of the estimation model. In contrast, VAEs only optimise a lower bound, which leads to a less identifiable model.

In this report, we perform an extensive study on both the reproducibility of the results presented in the paper as well as the supposed benefits of using iFlow over existing representation learning methods as claimed by the authors [7]. The contribution of this work is therefore two-fold: We reproduce original results presented in paper using the existing implementation, add some baseline experiments to better place the obtained results into context, and discuss the metric used for evaluation. Next, we analyse the claimed practicality of the proposed method and perform experiments to gauge the benefits that identifiability brings to learning stronger, and more importantly, explainable latent representations. We provide the source code[1] to run the experiments performed in this report.

## 1.1   Target Questions

To verify the results presented in the paper and to asses the claimed superiority of iFlow over previous deep generative models on representation learning tasks, this report mainly focuses on answering the following four questions:

- Can the claimed superiority of iFlow over iVAE with respect to MCC-scores be reproduced using only the methods described in the paper and provided source code?
- How well does iFlow perform compared to its non-identifiable counterparts, considering multiple flow-based models such as Spline-Flow and Planar-Flow?
- To what extent does iFlow offer practical advantages over existing deep representation learning frameworks such as ICA, iVAE and Flow?

---

[1]A link to our code-base will be made available in the camera-ready version of our paper.

Submitted to 34th Conference on Neural Information Processing Systems (NeurIPS 2020). Do not distribute.

81 • To what extent does identifiability improve upon normalising flows?

## 2 Reproducibility of Mean Correlation Coefficients

83 Recently, [5] proposed an identifiable Variational Auto Encoder (iVAE) that reaches identifiability by
84 conditioning the latent variables on an auxiliary variable. However, the authors of [7] claim that iVAE
85 leads to sub-optimal identifiability, both in theory and practice. This is due to the intractable KL
86 divergence between the approximate and true posterior and iVAE only maximising the lower bound
87 log-likelihood evidence. To overcome these limitations, the authors propose identifiable generative
88 model through flows (iFlow). According to the authors, its ability to directly maximise the likelihood
89 yields stronger identifiable latent representations, achieving better principled disentanglement.

90 In this section, we reproduce the original results as presented by the authors. We will evaluate
91 using the implementation provided by the authors. Specifically, we reproduce the Mean Correlation
92 Coefficient (MCC) scores, which is a standard measure used in independent component literature.
93 As the MCC metric can be sensitive to synthesised data for different seeds, the authors run the
94 experiment for 100 different data seeds. Due to time constraints, we run our experiments for 20 data
95 seeds, which is ample to verify the original results. We use the same Gaussian time-series dataset as
96 used in the paper, as described in [3] (see Supplemental Material Section B). For the flow function,
97 the authors use RQ-NSF(AR) cubic-spline flow [2]. All the relevant hyper parameters and hardware
98 configurations can be found in Supplementary Materials Section A.1.

99 It must be noted that there is some discrepancy between the paper and the provided implementation.
100 In the paper, it is mentioned that the softplus non-linearity is applied to the final layer of the auxiliary
101 mapping function $\lambda$. To ensure finiteness, a negative activation is then taken over the second order
102 parameters. In the implementation, however, it is suggested that the softplus non-linearity should only
103 be applied to the second order parameters. Therefore, we evaluate both methods. The different models
104 used throughout the experiments are implemented using the PyTorch Deep Learning Framework [8].

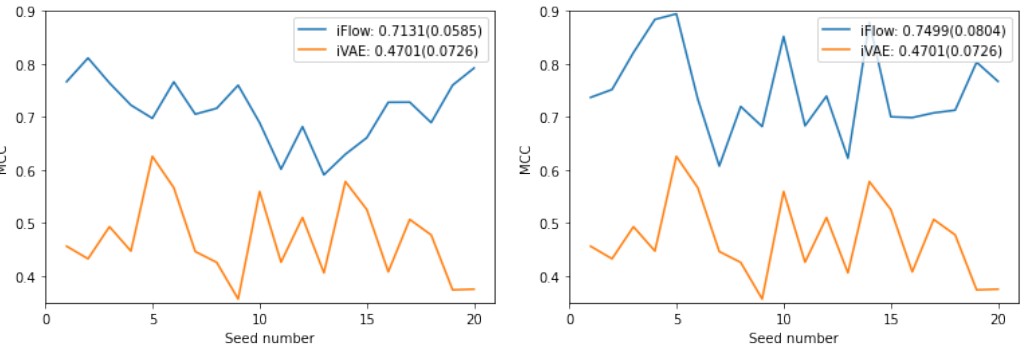

Figure 1: **Left**: results using the implementation as described in the paper. **Right**: results using the implementation as suggested in the code-base. Legend includes mean MCC, and standard deviation in parenthesis.

105 Figure 1 shows the MCC scores achieved by iFlow and iVAE for both the original implementation
106 as described in the paper and the alternative implementation suggested in the code-base. When
107 comparing our results to those presented in the original paper, we observe merely small differences.
108 For the original implementation, our experiment yields slightly lower results for iFlow for the mean
109 MCC and standard deviation with a difference of 0.011 and 0.005 respectively. For iVAE, our mean
110 MCC scores 0.0263 below the result of the paper, and the standard deviation is more or less equal.
111 For the alternative implementation, we reach an mean MCC score higher than both our previous
112 result and those presented in the paper, with a difference of 0.0258 for the latter. We also observe a
113 slightly higher standard deviation. Our results do not make clear which of the two implementation is
114 correct, as the differences between the results are minor.

115 Both our results and those presented in the paper show a relatively high standard deviation over
116 different data seeds, while net network seed remains fixed. Therefore, we conduct a further experiment
117 where we compare a fixed network and variable data seed with a variable network and fixed data

seed for both iFlow and iVAE to gauge model stability. Again, we evaluate on 20 different seeds for each experiment. For iFlow, Figure 2 shows very comparable mean MCC scores and standard deviations when fixing either the data or network seed. This suggests the seed chosen to run the experiments in the paper was not cherry-picked. For iVAE, we observe very similar mean MCC scores, but the standard deviation is halved when the data seed is fixed. This may be attributed to having a particularly well suited data seed for this specific case. In conclusion, our results seems to be in line with those presented in the paper, also when considering variable network seeds. We observe that iFlow indeed performs significantly better than iVAE, verifying the authors' claim.

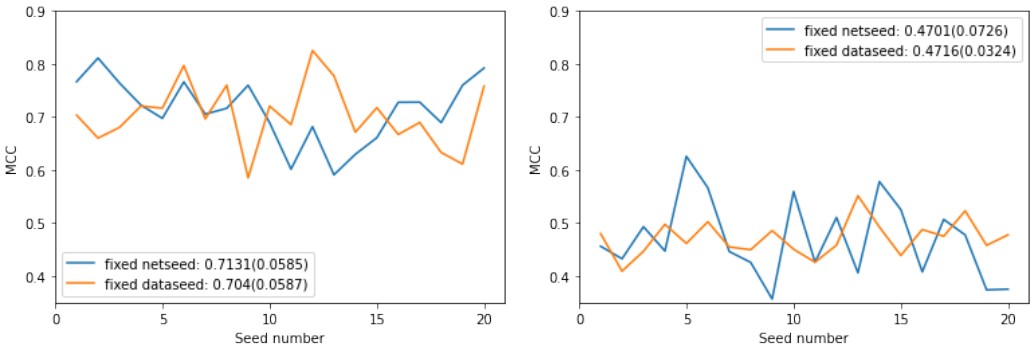

Figure 2: **Left**: results for iFlow. **Right**: results for iVAE. Legend includes mean MCC, and standard deviation in parenthesis.

# 3 Comparing iFlow to Flow

As is shown in Section 2, our findings are in line with those that are reported in the original paper [7]. The results show that iFlow reaches higher MCC scores compared to iVAE. However, these results are not placed into context: Two inherently different models are being compared. Therefore, we conduct an experiment where we compare the identifiable flow models with their non-identifiable counterparts. It is important to note the difference in parameter complexity between iFlow and iVAE. The authors use the Q-NSF(AR) cubic-spline flow [2], making their iFlow implementation contain 2.944.980 compared to 18.170 parameters for iVAE. Furthermore, the original code base also includes Planar-Flow model containing 2.170 parameters. We evaluate MCC-scores for both flow models. The result are displayed in Figure 3 and include iVAE performance for reference.

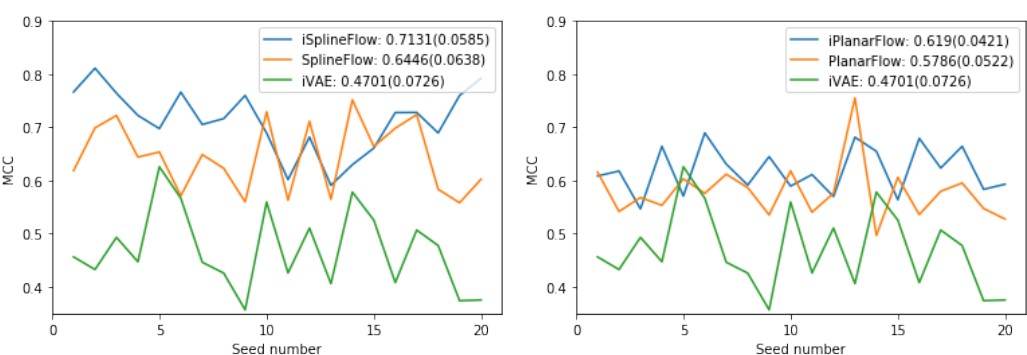

Figure 3: **Right**: results for Spline-Flow. **Left**: results for Planar-Flow. Legend includes mean MCC, and standard deviation in parenthesis.

We immediately observe that both non-identifiable flow models reach significantly higher MCC scores compared to iVAE. Clearly, both flow-based models allow for learning a richer representation. We also observe that the spline-based model outperforms the planar-flow model, likely due to its increased parameter count. Interestingly, we do notice slightly higher mean MCC-scores for the identifiable flow models. However, since performance is measured using MCC, these difference are

slim, and seem to suggest a slight improvement at best, especially for the planar-flow model. This is something the authors fail to mention, but we feel it is an important aspect to gain a better perspective on the actual benefit of using iFlow.

We also want to raise two important observations regarding the MCC metric used to evaluate the performance of the identifiable models. First of all, it must be noted that the discussed identifiable models are described as being identifiable up to some equivalence class. Specifically, they are identifiable up to some invertible affine transformation over the sufficient statistics. This does not seem to be accounted for when the MCC is calculated between the original sources and learned latent representation.

Moreover, we ran a small experiment where we replaced the flow mapping function to an identity function. With this model, we measured a significant correlation between the data samples and sources from which these samples are generated. For the data used in the experiments described in Section 2, we found an MCC score of 0.58 between data and sources. This means that the model often learns a latent representation that is less correlated with the original sources compared to the data itself. Note that in general the MCC metric varies greatly between both different data seeds and different network seeds because the MCC metric can be quite susceptible to small deviations in the learned latent representations. Because of these reasons, we question the effectiveness of the MCC metric to evaluate the model's ability for learning a true latent representation.

# 4 Assessing iFlow's Usability

In this section, we clarify to what extent iFlow offers practical advantages over previous representation learning methods. In the paper [7], the authors specifically mention three existing representation learning frameworks: Nonlinear independent component analysis (ICA) [4], identifiable Variational Auto-Encoders (iVAEs) [5], and Flow-based models [6]. The authors explicitly ascribe all three frameworks specific shortcomings and claim that by adding identifiability to flow-based models, iFlow provides practical advantages over existing methods. However, besides expanding on iFlow's theoretical ability to recover the true latent space, and showcasing that iFlow achieves higher MCC-scores (Section 2) than iVAE on synthesized Gaussian time-series data [3], the authors do not explicitly make clear to what extent iFlow overcomes or preserves other mentioned shortcomings of ICA, iVAE and Flow. In this section, we elaborate on iFlow's practical usability as compared to these previous methods by investigating how readily iFlow can be used on various datasets.

We explored iFlow's practical usability by attempting to apply it to two datasets: the MNIST dataset and the Gaussian Blob dataset. We chose these two datasets because MNIST is one of the most frequently used datasets in machine learning research in general, and because the Gaussian Blob dataset is frequently used particularly in representation learning research.[2] For a more elaborate explanation of the datasets, see Figure S3 and Figure S4 in the supplementary materials. For both datasets, we took the source code for iFlow and tried to re-implement it such that iFlow should be able to learn the dataset's latent distribution. Our findings are summarized in Table 1.

Table 1: Overview of iFlow's practical usability compared to alternative representation learning methods.

|  | Requires Auxiliary Variable | Identifiable | Latent Space Distribution | Allows for Sampling | Allows for Dimensionality Reduction |
|---|---|---|---|---|---|
| ICA | Yes | Yes | N/A | N/A | Assumed Not |
| iVAE | Yes | Yes | Approximation | Yes | Yes |
| iFlow | Yes | Yes | Exact | Yes [3] | No [4] |
| Flow | No | No | Exact | Yes | No [4] |

For MNIST, we found that the provided labels (i.e. numbers) could function as auxiliary variables. Nevertheless, both in the paper as well as in the provided source code, the dimension of the observed

---

[2]The Gaussian Blob dataset is also used by the $\beta$-VAE paper to show its principled disentanglement.

datapoints and the latent space must be of the same size for iFlow to work. [4] However, in MNIST datapoints are of size 28 by 28, and it is very unlikely that that the underlying latent space is of size 784: That would imply that each pixel is an explanatory source in itself, while we reason that there should be no more explanatory sources for recognizing single numbers than that there are single numbers to begin with. iVAE does not have this limitation: There actually is an implementation iVAE for MNIST available. Furthermore, in contrast to the synthesizes Gaussian data used in the paper, we do not know beforehand what the latent space of MNIST should look like. It is therefore difficult to evaluate how well iFlow recovers this latent space because we cannot compare it to the original latent space; making it impossible, for instance, to compute MCC-scores.

In contrast to MNIST, for the Gaussian Blob dataset the latent space is known beforehand, potentially making it a candidate for evaluating iFlow's performance. However, the constraints iFlow puts on the dimensions of the observed and latent space are again problematic: because the datapoints are again images, iFlow's assumption that the latent space is of the same size as the observed datapoints seems once again to result in a latent space with much more dimensions that we should reasonably expect. Furthermore, it is not straightforward to determine what should be used as the auxiliary variable for this dataset. We could use the quadrant indices indicating in which quadrant the blob appears as auxiliary variable, but this is only possible because this dataset, just like synthesized Gaussian Time-series data, is generated artificially and we therefore know the underlying distribution.

In conclusion, our exploration of attempting to apply iFlow to different datasets shows that iFlow practical usability has more caveats than warranted by the authors. First of all, we assume the dimensionality of latent space to be (much) smaller than dimensionality of the observed datapoints. Nevertheless, iFlow's current implementation requires the latent space to be of the same size as the observed datapoints, leading to unreasonably high-dimensional latent spaces when working with pixel-valued image data. Furthermore, using datasets where the underlying distribution is not known beforehand (which can be reasonably be expected to be the most common case for practical applications) makes it non-trivial to evaluate iFlow's performance and to come up with the required auxiliary variable. While this requirement for an auxiliary variable is inherent to all identifiable methods, this does put severe limits on the applicability of these models. Consequently, non-identifiable models such as Flow can be more readily applied to a much wider range of datasets. Why then, one would consider using iFlow at all, we discuss in Section 5.

# 5 Evaluating iFlow's Disentangled Identifiability

In addition to the supposed practical advantages iFlow has over other methods, which we have nuanced in Section 4, the authors mainly emphasize iFlow's theoretical guarantee for learning the true latent representation. Furthermore, while the authors do provide visualisations of learned latent representations, these visualisations do not lend themselves well for identifying whether the model has learned a principled disentangled representation. In this section, we aim to further analyse iFlow's recovered latent representation, and how it improves upon the representation recovered by its non-identifiable counterpart Flow. First, we provide a brief overview of the notion of entanglement.

Disentangled representations differ from entangled ones in at least one significant way. Both entangled as well as disentangled representations help leverage a significant bottleneck conventionally present in machine learning work: instead of having to labor-intensively use human ingenuity to feature-engineer the representations that support effective machine-learning, representation learning frameworks use available data to learn these representations in an unsupervised-manner [1]. Disentangled representations not only learn representations that lead to discriminative information for the task at hand, but they also learn to successfully separate the various explanatory sources underlying the data[5]

---

[3]Theoretically iFlow should allow for sampling, but the source code provides no such functionality.

[4]Theoretically, we should be able to circumvent this limitation of iFlow by engineering the latent space of the underlying flow-based model in a very task-specific way, but we found no code or suggestions in the paper to do so.

[5]For example, when we look at a glass on a table, in our mind we can very readily separate the glass from the table, and even from the shadows it casts on that table. However, the stimuli of all three sources reach our retina in the same way. Apparently, something in our brain learns to separate these various sources that explain the stimuli we receive. A disentangled representation learning framework aims to achieve the same for machine learning models that receive input data.

[1]. Principled disentangled representations go a step further: not only are these explanatory sources successfully separated, the components modeling these sources are independent.

To investigate to what extent iFlow leads to a more principle disentangled learned representation than Flow, we used the provided source code to implement an identifiable RealNVP model. From here on, we refer to RealNVP as Flow, and the identifiable RealNVP as iFlow. We evaluated both models on the Half Moon dataset and compared their respective learned representations. The reasons for choosing this dataset are twofold: First of all, both Flow and iFlow can be readily applied to this dataset because the dataset has a well-defined auxiliary variable and has none of the dimensionality-problems as described in Section 4. Secondly and more importantly, the dataset has a latent space that is relatively easy to comprehend and therefore allows us to manually evaluate to what extent both frameworks entangle or disentangle the learned latent representation. For more details on the dataset, see Figure S2 in the supplementary materials.

The paper and provided source-code do not offer a method to generate new datapoints using iFlow. We found that coming up with a own sampling technique is not straightforward, because nowhere during the forward pass of the model is the latent space saved for later use. Nevertheless, it seems theoretically possible to design and implement an additional deep model that learns this encoding step. However, due to time-constraints, we choose to model the data as a simple two-dimensional multivariate Gaussian (tuning $\mu$ and $\sigma$ on the data). For Flow, the data is generated using the given prior and is classified using the nearest neighbour algorithm and previously observed data. The results are shown in Figure 4.

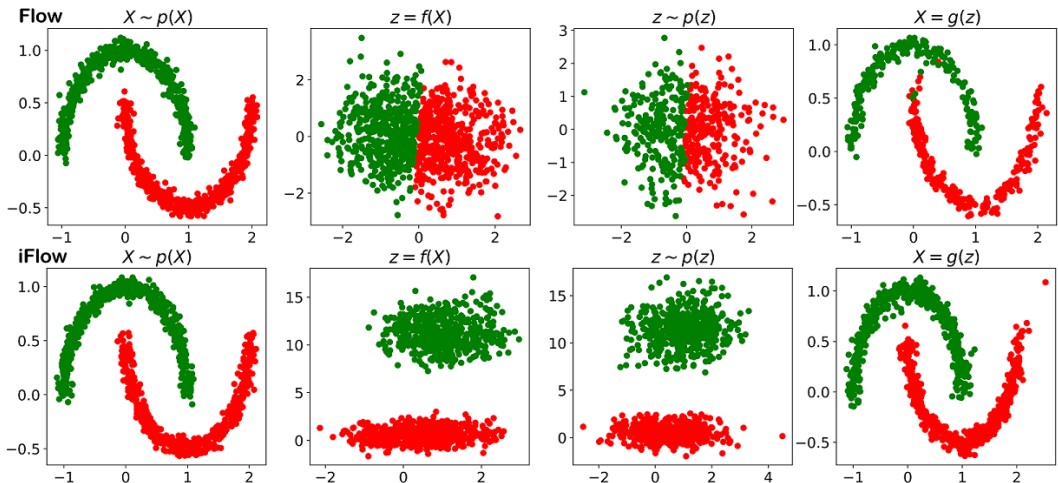

Figure 4: **Top row**: results for Flow. **Bottom row**: results for iFlow. From left to right: data samples, learned latent representation, estimated latent density function, generated samples. Note that for visualisation purposes we randomly sampled 1000 points from the used dataset. These samples were not used for training.

The results seem to show that iFlow indeed leads to slightly better disentanglement than Flow: iFlow more distinctly separates both group of datapoints into independent clusters. Consequently, generated datapoints using iFlow also seem to more accurately map back onto the originally observed distribution. However, for principled disentanglement we should also check whether moving along one axis within the recovered latent clusters only results in variation along one underlying explanatory source while keeping other sources relatively constant. To investigate this, we ran one final experiment for both trained models.

Ideally, we want to observe the mapped latent space in a continuous and global way, where each axis corresponds to an independent property of observed data. This would suggest the model has learned a principled disentangled latent representation. Figure 5 shows the mapping of the well-defined region of the learned latent representation to the original data space. Here, we segmented the latent space using different icons to distinguish the gradient of the space in the horizontal direction and colours for the gradient in the vertical direction.

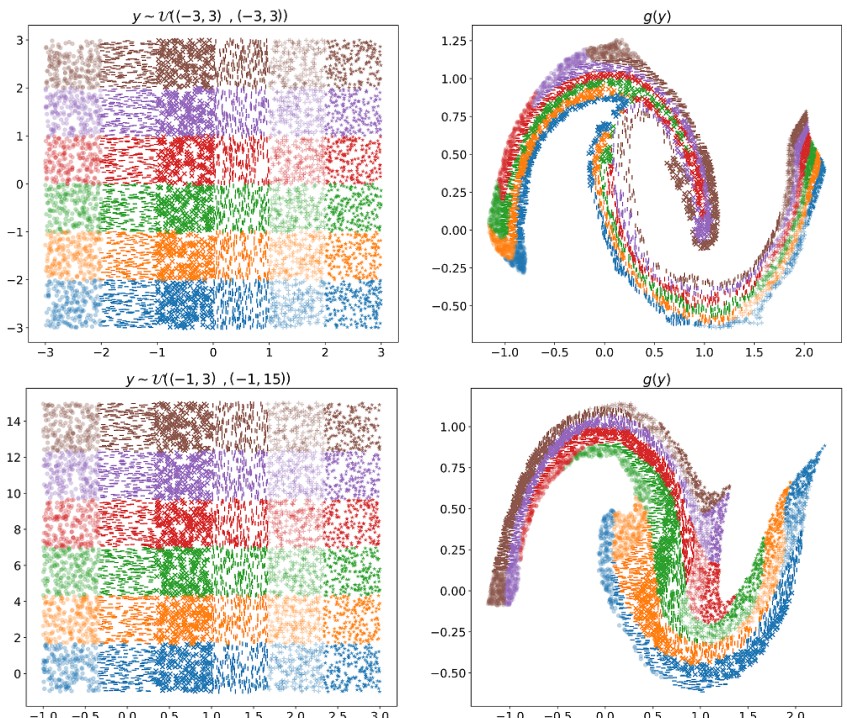

Figure 5: **Top**: results for Flow. **Bottom**: results for iFlow. From left to right: learned latent representation and the mapping of this representation to data space. To gauge latent gradients different icons and colours are used for the horizontal and vertical components respectively.

In Figure 5 we can indeed observe the desired properties for the identifiable model. The meaning of the latent space is globally defined. When we move across the horizontal dimension while fixing the vertical dimension, a clear gradient across the original half moons is present. The vertical position indicates which half moon the latent space is mapped to. For the non-identifiable model, this is less defined. Here, the horizontal dimension encodes both the position on each half moon and to which half moon a point is mapped to. The vertical dimension only encodes the vertical position of a point on the half moons.

However, it must be noted that the results presented in Figure 5 are not guaranteed. First, we observe that for both Flow and iFlow the meaning of the latent space is not always globally defined, although local meaning may still be present, as in Figure S7c. Also, the identifiable model often does not reach the claimed principle disentanglement, showing a learned latent representation that is similar to Flow which can be observed in Figure S7a. One key difference that is consistently observed between Flow and iFlow, the the ability to learn a continuous latent representation. It seems that for iFlow, the relative structures in the learned latent representation remain intact, whereas for regular Flow, this does not have to be the case. This is shown in Figure S8b. These preliminary findings indicate that iFlow can indeed be effective in learning a stronger latent representation compared to Flow.

## 6  Conclusion

In this work, we investigated the reproducibility of the results presented in [7], comparing iFlow's performance in recovering a true latent representation to the previously proposed iVAE. Next, we elaborated on the fairness of the metrics used in the paper. Finally, we assessed the usability and practical advantages the authors claim iFlow has over existing representation learning methods.

We found that we could reproduce the reported MCC-scores of iFlow and iVAE to acceptable precision, having us conclude that the results in the paper showing that iFlow outperforms iVAE with respect to these MCC-scores are valid. However, we found that the paper does not put these results into context. Therefore, we investigated iFlow's performance when replacing its underlying cubic-spline flow model with a less parameter-intensive planar-flow model. We found that while planar-based iFlow still outperforms iVAE with respect to MCC-scores, it does so by a significant lesser degree. Furthermore, we show that their non-identifiable counterparts outperform iVAE as well, and the difference in MCC-scores between the identifiable and non-identifiable models is less profound when compared to the difference between iFlow and iVAE.

Next, we found a significant correlation between the data and the underlying distributions from which this data is generated. This raises the question whether MCC-scores alone are sufficient and valid in determining whether the true latent representation is found. We also attempted to apply iFlow to recover the latent representations of two datasets not presented in the paper: The MNIST dataset and the Gaussian Blob dataset. We found that mainly iFlow's requirement for auxiliary variables and specifically sized input data puts significant constraints on datasets it can be used for, having us conclude that the authors claim for iFlow's practical advantages over other existing methods seems too simplistic.

Finally, we evaluated how well iFlow delivers on its promise for principled disentanglement as compared to its more readily applicable, but non-identifiable, counterpart Flow. By visually exploring the learned representations of both models on the half-moon dataset, we found that iFlow is capable of recovering principled disentangled representations, where Flow is not. However, often this state of representation is not reached, and the obtained representation is similar to that of Flow. Nevertheless, we do observe a consistent stronger recovered latent representation when utilising iFlow.

In conclusion, iFlow shows us a promising next step in the area of recovering true latent representations and reaching principled entanglement. Future research in identifiable models and possible metrics to further analyse their performance is definitely warranted, as this could solve many of the issues illustrated in this reputability report.

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

# Supplementary Materials: [Re] Identifying Through Flows for Recovering Latent Representations

## A Hyperparameters and Hardware for Conducted Experiments

### A.1 Original Paper Experiments

Table S1: Hardware configurations used to train the various models. Time indicates the amount of minutes required to fully train a single model.

| Model | GPU | CPU | RAM | Time (minutes) |
|---|---|---|---|---|
| (i)Flow (spline) | RTX 2080 Ti | - | 32 GB DDR4 | 20 |
| (i)Flow (planar) | RTX 2080 Ti | - | 32 GB DDR4 | 7 |
| iVAE | RTX 2080 Ti | - | 32 GB DDR4 | 1.25 |
| (i)RealNVP | - | AMD 3900X | 32 GB DDR4 | 3 |

Table S2: Overview of hyper-parameters for the models used in the reproducibility paper.

| | iFlow (Spline) | iFlow (Planar) | iVAE | RealNVP | iRealNVP |
|---|---|---|---|---|---|
| lr (ADAM) | 0.001 | 0.001 | 0.001 | 0.0001 | 0.0001 |
| scheduler | on plateau | on plateau | on plateau | after $10^4$ steps | after $10^4$ steps |
| lr drop factor | 0.25 | 0.25 | 0.25 | 0.1 | 0.1 |
| lr patience | 10 | 10 | 10 | - | - |
| batch size | 64 | 64 | 64 | 64 | 64 |
| epochs | 20 | 20 | 20 | 1 | 1 |
| steps/epoch | 625 | 625 | 625 | 15000 | 15000 |
| lambda MLP dims | 40, 30, 20, 10 | 40, 30, 20, 10 | - | - | 2, 8, 4 |
| lambda activation | ReLU + SoftPlus | ReLU + SoftPlus | - | - | ReLU + SoftPlus |
| iVAE architecture | - | - | MLP (5,50,50,5) leaky ReLU | - | - |
| number of bins | 8 | - | - | - | - |
| flow length | 10 | 10 | - | 3x2 | 3x2 |
| coupling layer architecture | - | - | - | MLP (2,256,256,2) leaky ReLU | MLP (2,256,256,2) leaky ReLU |

 # B   Datasets Explanation

 ## B.1   TLC Dataset

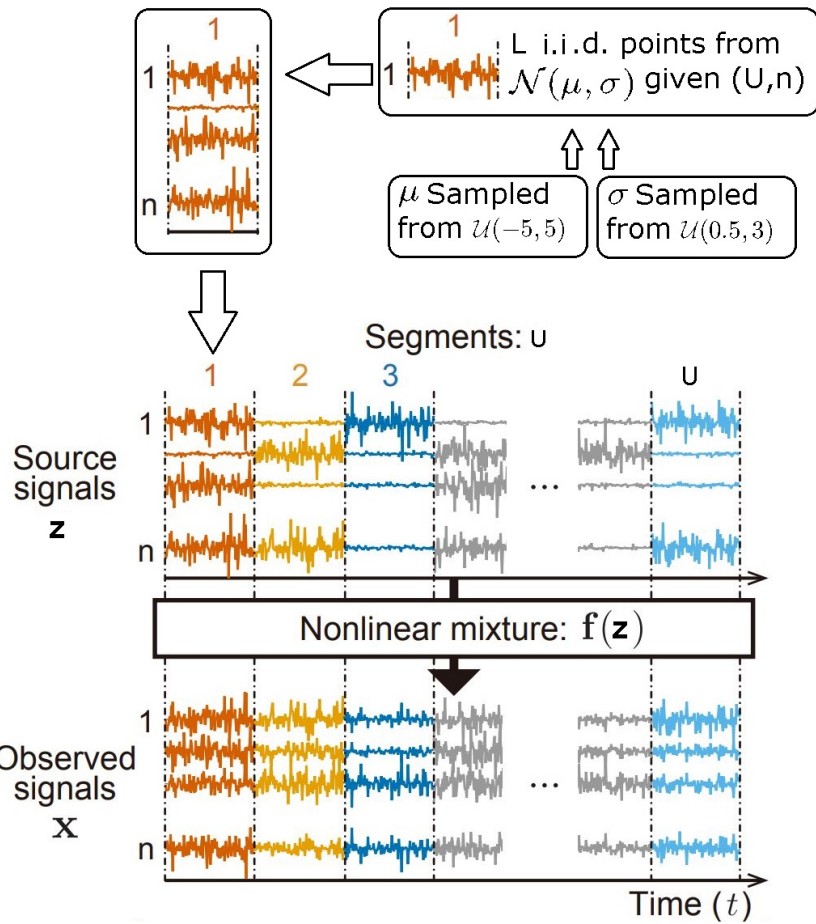

Figure S1: A visualization of the Gaussian time-series dataset consisting of U labels/segments of L i.i.d. points [1]. Each source signal is an 5-dimensional vector sampled from a multivariate Gaussian $\mathcal{N}(\mu, \sigma|U)$, where $\mu \in \mathbb{R}^5$ is sampled from a uniform distribution $\mathcal{U}(-5, 5)$ and $\sigma \in \mathbb{R}^5$ is sampled from $\mathcal{U}(0.5, 3)$. X is obtained by f(z), 3 non-linear transformations with xtanh activation of the source signal z. Time (t) is not used to create this synthetic data.

 **B.2 Half Moon Dataset**

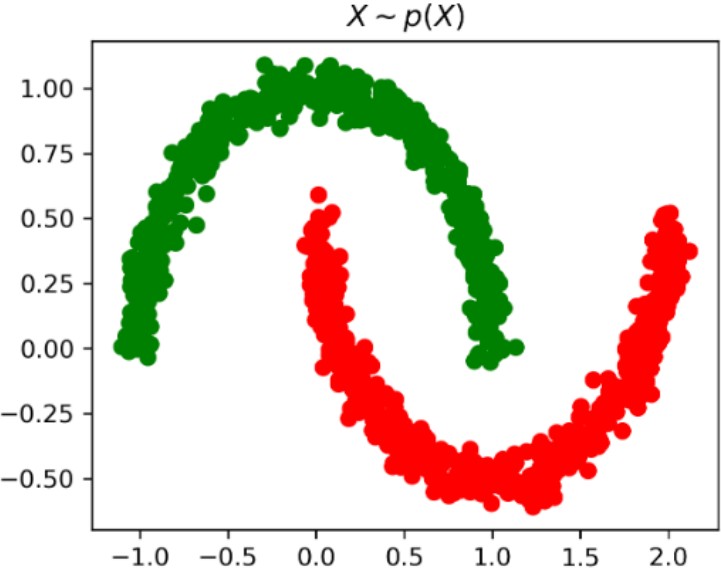

Figure S2: A visualization of the half moon dataset with two labels. The data, X, consists of green points labeled zero which are generated from the top half of a circle with centre $(0, 0)$ and red points labeled one which are generated from the bottom half of a circle with centre $(0.5, 1)$. [2]

 **B.3 MNIST Dataset**

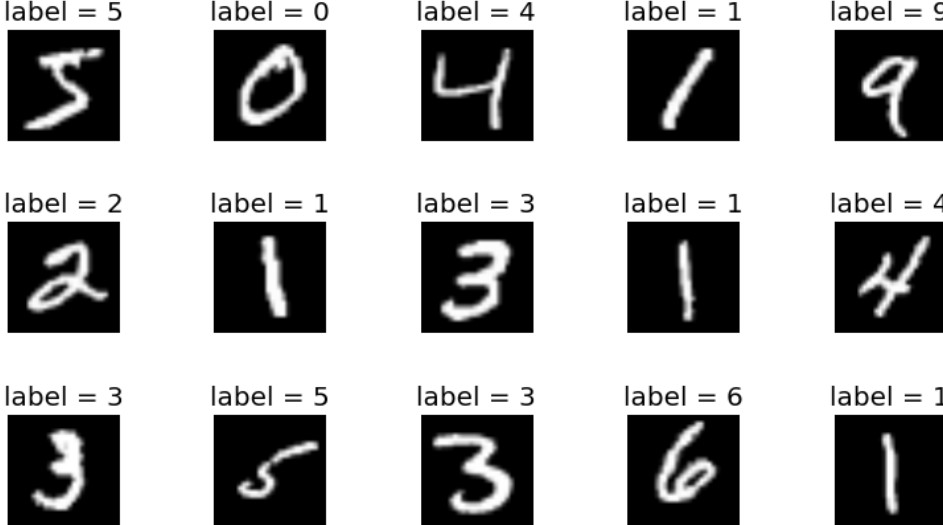

Figure S3: Examples from the MNIST dataset. The labels can be used as auxiliary variables.

 **B.4    Gaussian Blob Dataset**

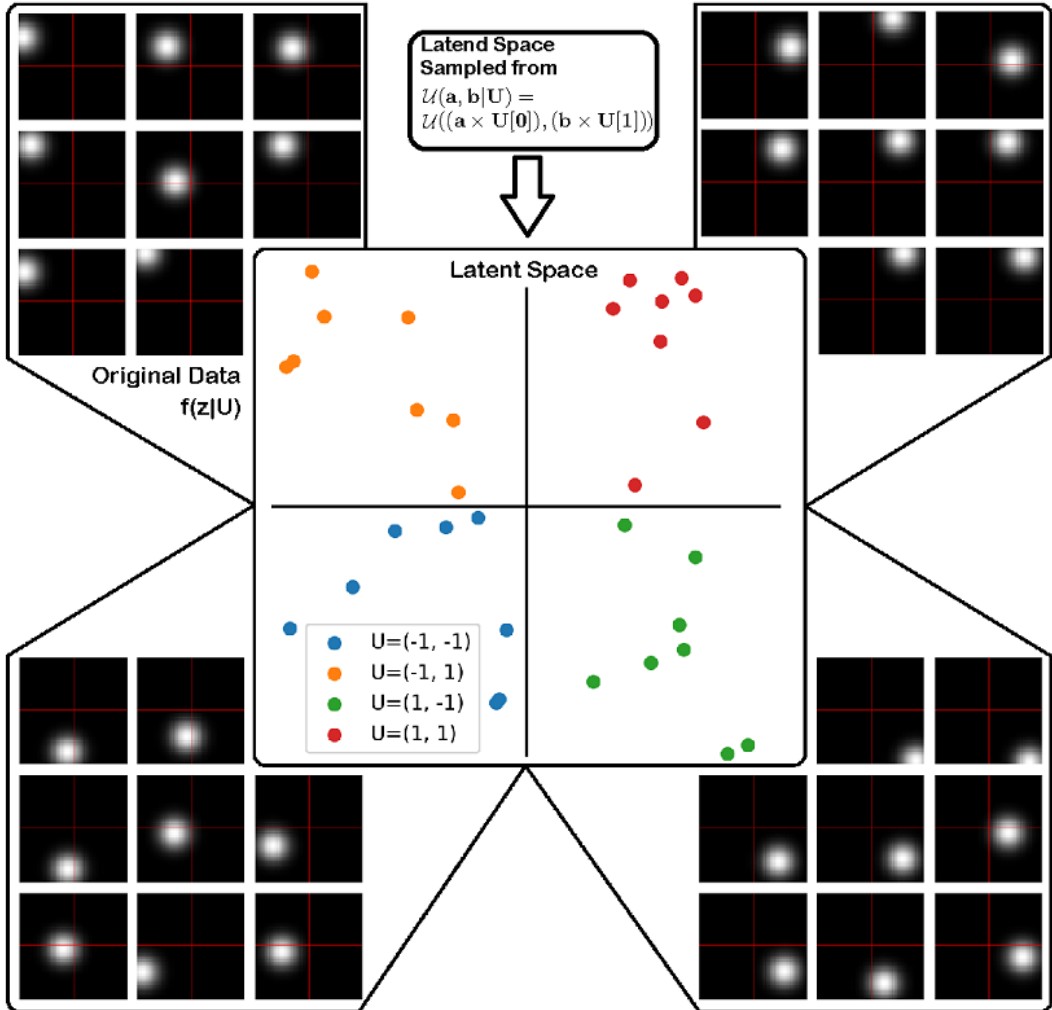

Figure S4: A possible implementation of auxiliary values for the Gaussian blob dataset. The centre of the figure represents the latent space coloured per auxiliary variable $U$. The four corners represent the original data created from the latent space given $U$. The data in each corner is created using a function $f(z|U)$ defined as a bi-variate Gaussian distribution with a identity covariance matrix and the latent space $z$ as the mean. It should be noted here that $U$ can only be created/used because the dataset is a synthetic dataset with a known latent space.

# C   Encoding, Decoding and Sampling for iFlow and Flow

## C.1   iFlow

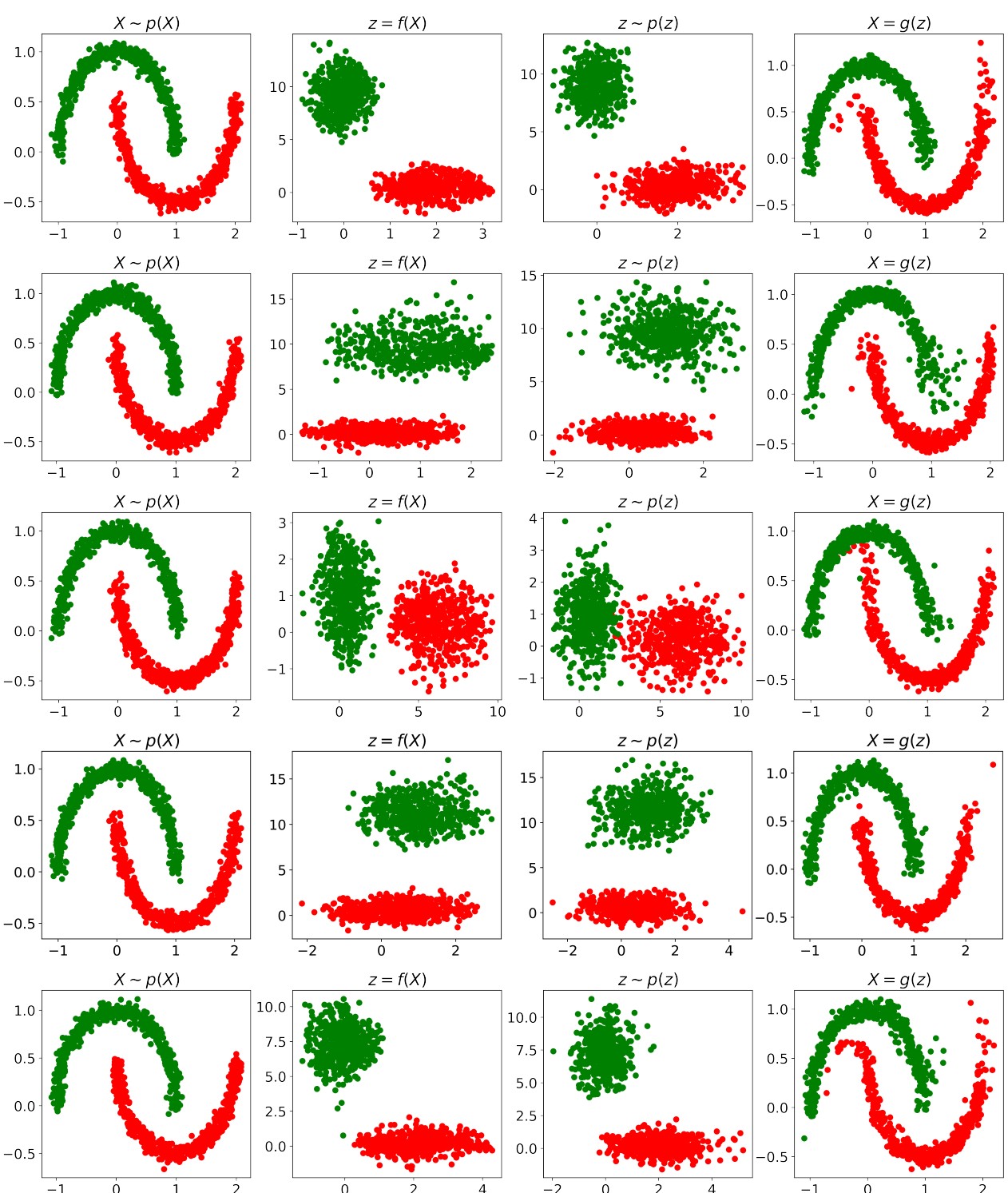

Figure S5: Five iFlow models (RealNVP) on the half moon dataset. The fourth model(top down) can also be found in the review. The models and the order in which they are placed are the same for the models in D.1. Left: original data ($X$), left middle: encoded data ($f(X)$), right middle: sampled latent data from the approximated Gaussian distributions ($z$), right: the decode sampled latent data ($g(z)$).

 **C.2 Flow**

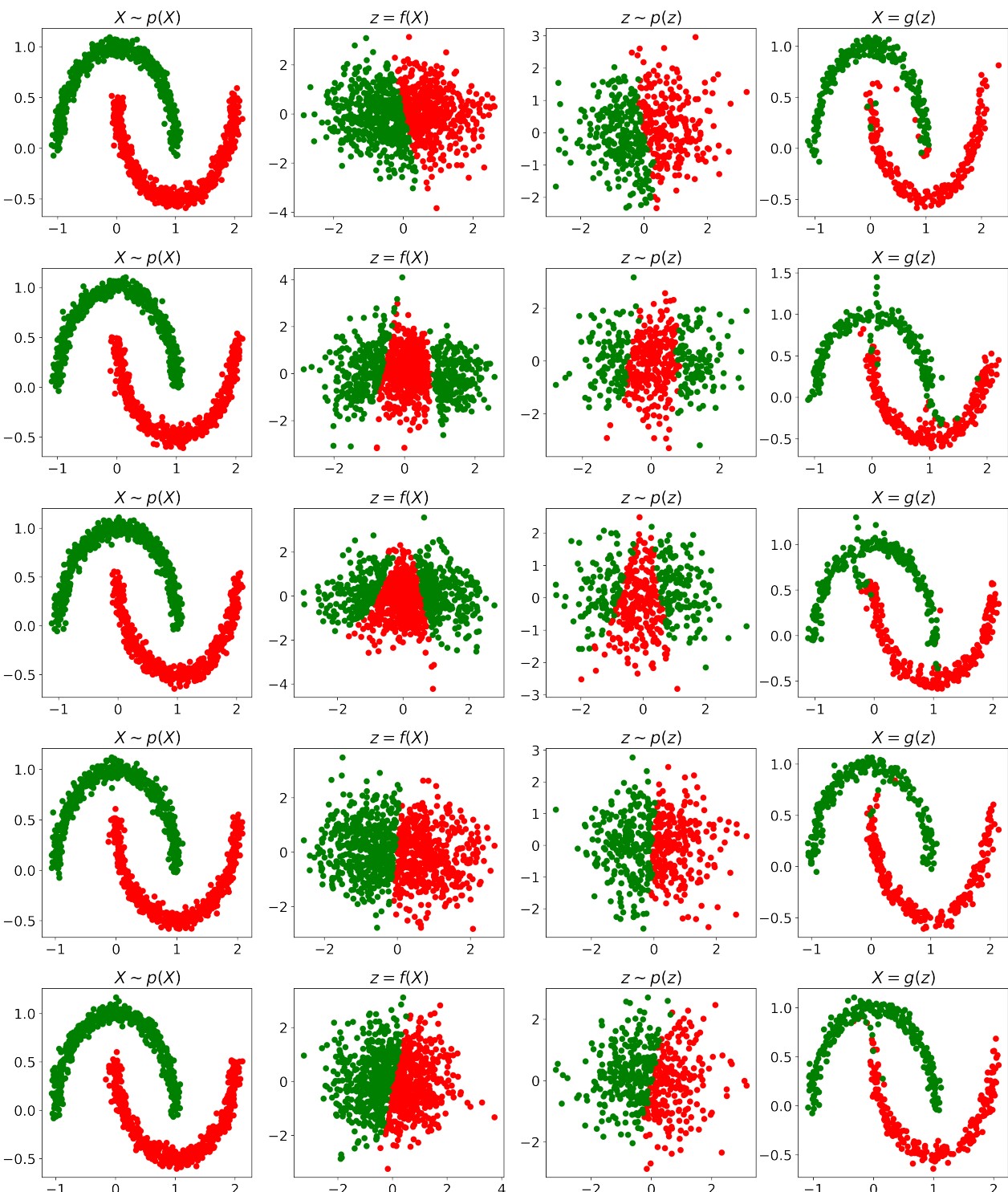

Figure S6: Five Flow models (RealNVP) on the half moon dataset. The fourth (top down) can also be found in the review. The models and the order in which they are placed are the same for the models in D.2. Left: original data $(X)$, left middle: encoded data $(f(X)$, right middle: sampled latent data from the Gaussian prior $(z)$, right: the decode sampled latent data $(g(z))$.

## D  Latent Space Exploration for iFlow and Flow

### D.1  iFlow

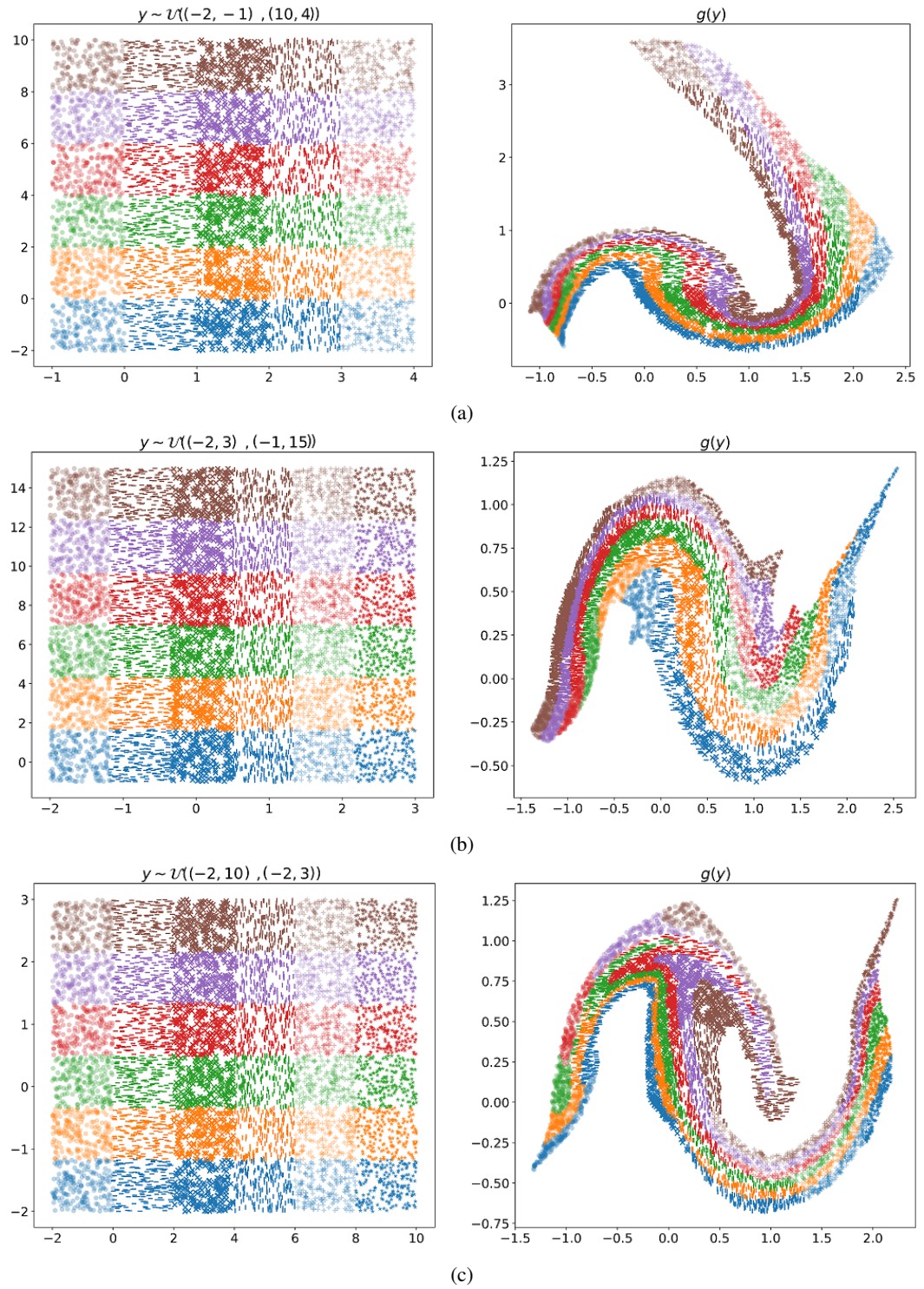

(a)

(b)

(c)

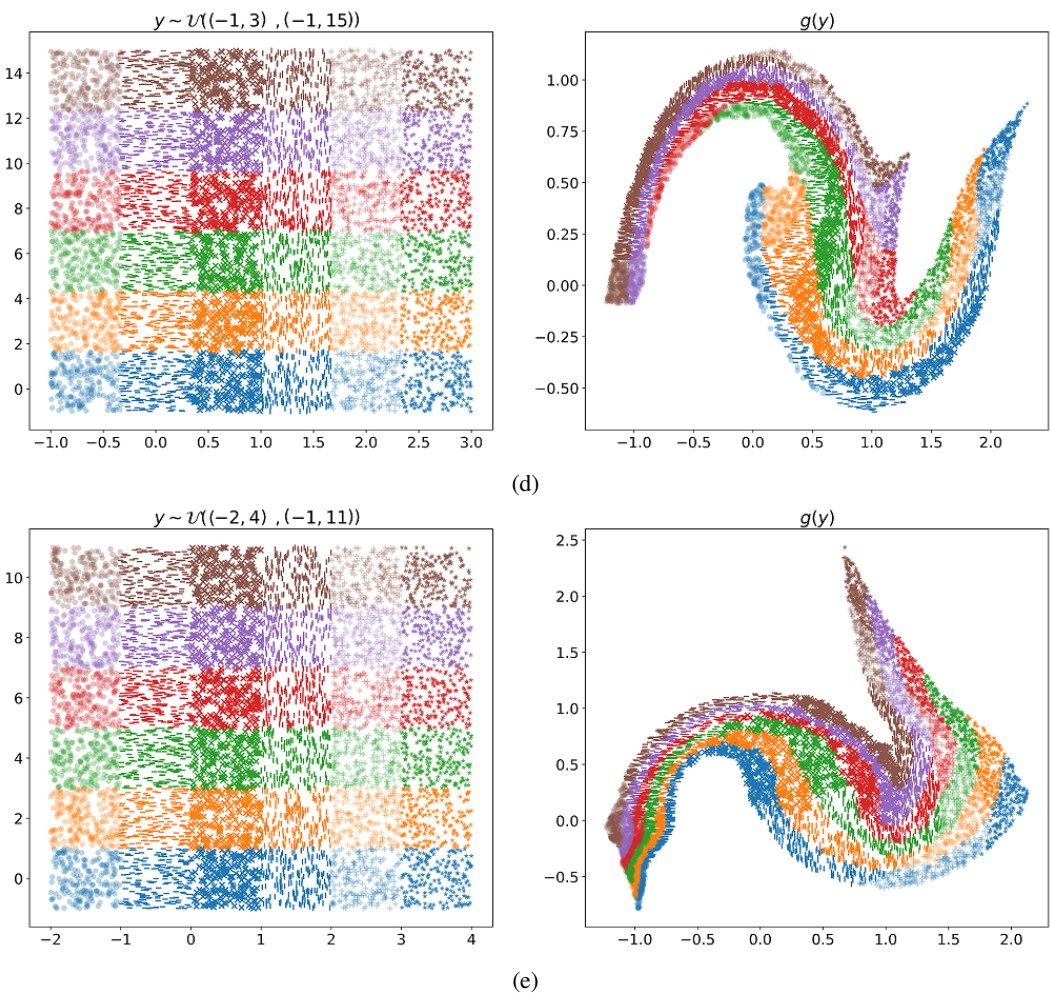

(d)

(e)

Figure S7: Five iFlow models (RealNVP) on the half moon dataset. The fourth (top down) can also be found in the review. The models and the order in which they are placed are the same for the models in C.2. Left: the latent space divide in a 6x6 grid, where each grid contains 200 samples uniformly sampled for each cell. Right: the original space, where the previous sampled latent space points are decode to the original space. Each cell has the same color and symbol in both spaces.

 **D.2 Flow**

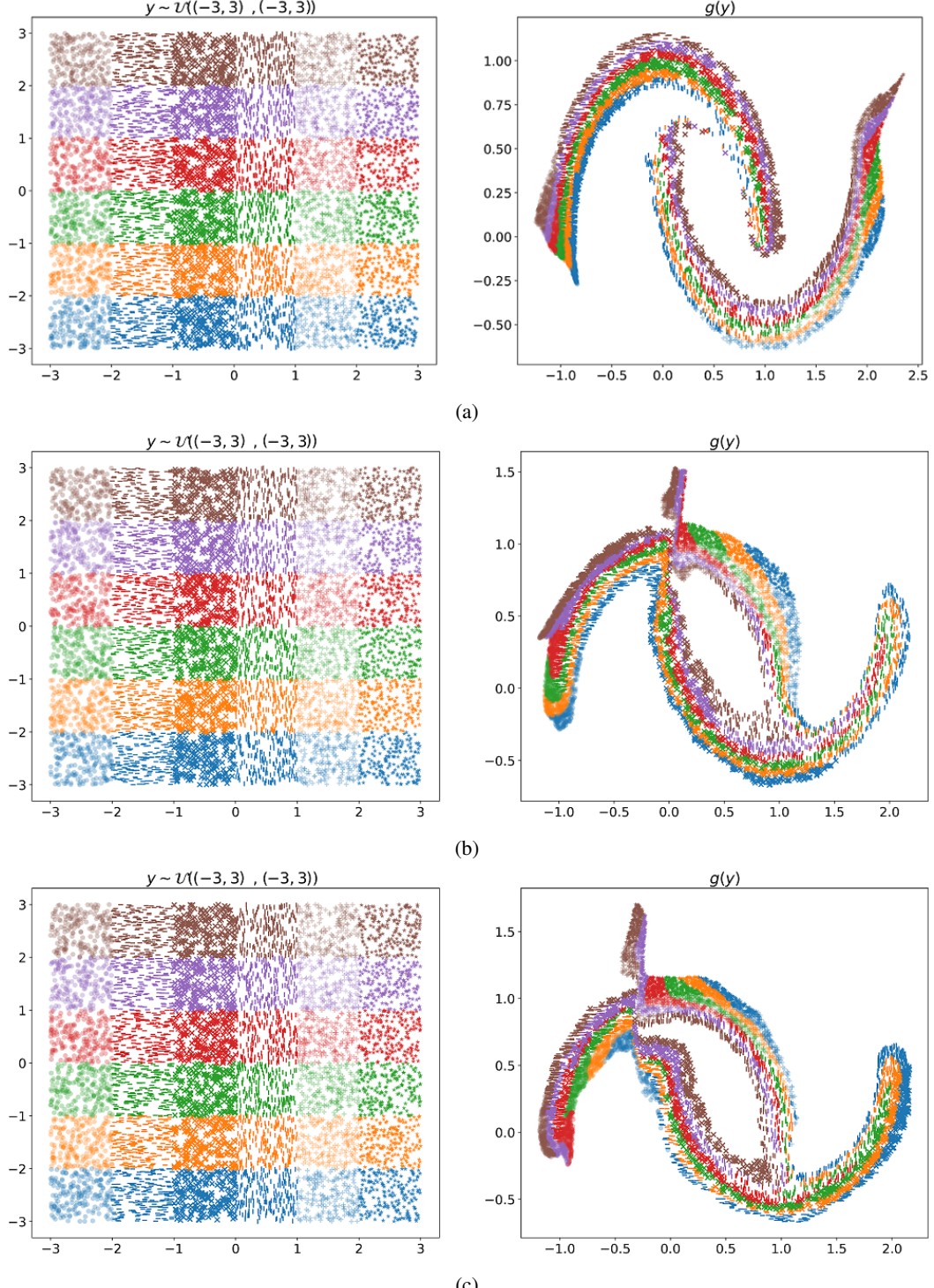

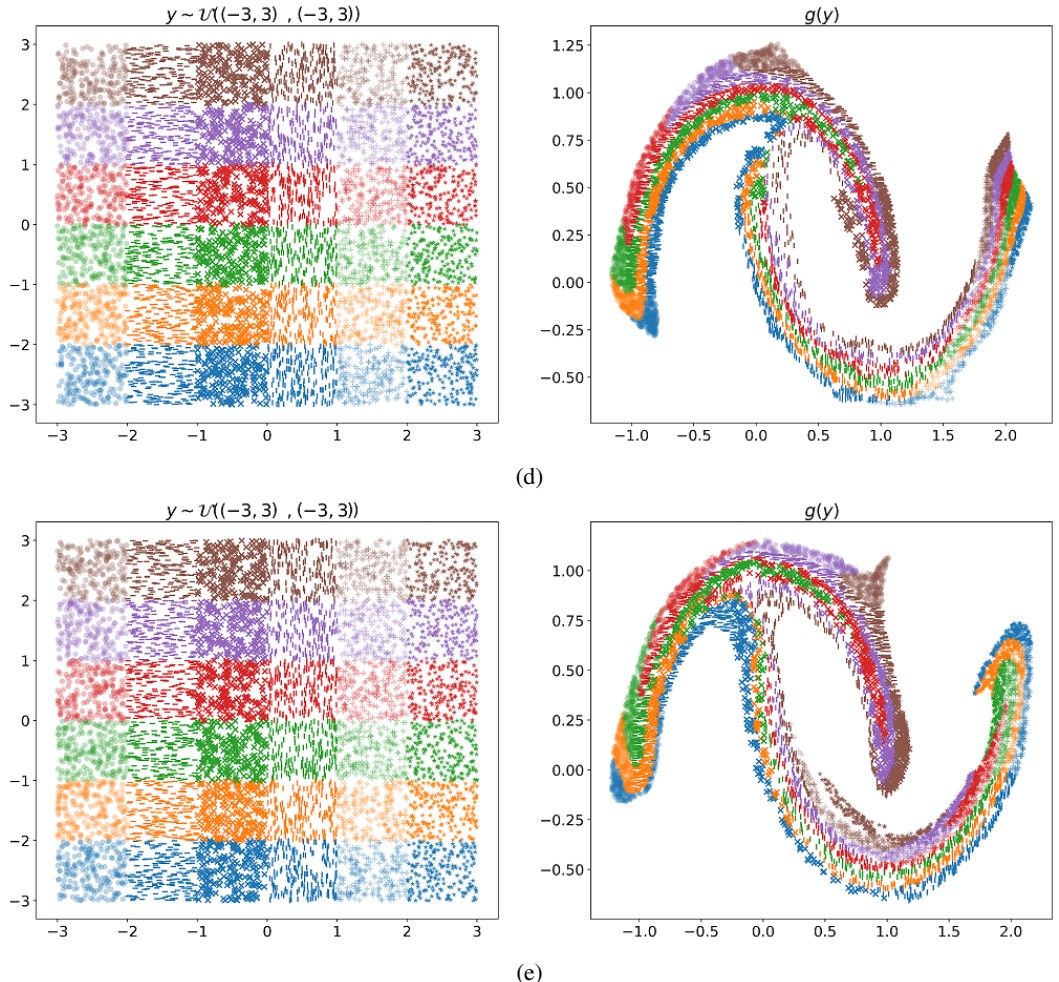

Figure S8: Five Flow models (RealNVP) on the half moon dataset. The fourth (top down) can also be found in the review. The models and the order in which they are placed are the same for the models in C.2. Left: the latent space divide in a 6x6 grid, where each grid contains 200 samples uniformly sampled for each cell. Right: the original space, where the previous sampled latent space points are decode to the original space. Each cell has the same color and symbol in both spaces.

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
