# OpenReview forum: "[Re] Identifying Through Flows for Recovering Latent Representations"
_ML_Reproducibility_Challenge/2020 — Reject_

### Official Review · AnonReviewer3 · 2021-03-01
**ML Reproducibility challenge: Identifying Through Flows for Recovering Latent Representations**

**Rating:** 7
**Confidence:** 2

**Review:**

This paper tries to reproduce the results provided in "Identifying Through Flows for Recovering Latent Representations". the authors reproduce the experiments in the original paper obtaining results, which are reasonably close to the original paper's reported numbers, diverging by only about 2.5%. The reproducibility report is written in a clear way having details of the experiments conducted. Furthermore, the authors investigate further experiments comparing the model with additional Flow-based models to give context to the devised model. Furthermore, further discussion is done on the applicability as well as the practicality of the iFlow model.

I have a few minor questions:
Why are the MCC results so different on different seeds? [Figures 1-3]
In figure 4, the data samples on which iFlow and Flow are compared look actually different and for the iFlow the data itself is more disentangled. Can you please provide some more comparable results for this?

**Familiar With The Original Paper:**

I have not read the original paper

**Reproducibility Summary:**

Report has summary

---

### Official Review · AnonReviewer1 · 2021-03-03
**extensive reproduction of the iFlow**

**Rating:** 7
**Confidence:** 4

**Review:**

This reproduction report does extensive works on different aspects to verify the original paper iFlow. The authors give a comprehensive study on three objects: 1) reproduce the results of MCC-scores presented in the original paper; 2) they elaborate on the fairness of the metrics used in the original paper; 3) they also evaluate the usability and practical advantages by working on different commonly used machine learning/presentation learning datasets. The authors of this report find some interesting points which are not presented in the original paper, and they also give a clear presentation of the experiments, analysis, and conclusion. Therefore, I feel this report is a good reproduction of the iFlow paper, which gives us a better and confidential understanding of the paper.

**Familiar With The Original Paper:**

I have not read the original paper

**Reproducibility Summary:**

Report has summary

---

### Decision · Program_Chairs · 2021-03-31

**Decision:**

Reject

**Comment:**

While the reproducibility effort is well attempted, it doesn't go above and beyond the reproduction and does not offer novel insights into the workings of the original paper.